# The Role of Autophagy in Pancreatic Cancer—Recent Advances

**DOI:** 10.3390/biology9010007

**Published:** 2019-12-28

**Authors:** Maria New, Sharon Tooze

**Affiliations:** Molecular Cell Biology of Autophagy Laboratory, The Francis Crick Institute, London NW1 1AT, UK; maria.a.new@outlook.com

**Keywords:** autophagy, cancer, PDAC, hypoxia

## Abstract

Pancreatic ductal adenocarcinoma (PDAC) remains one of the deadliest cancers with a 5-year survival rate of only 9%, despite ongoing efforts to improve treatment. This dismal prognosis is due to the difficulty of early stage diagnosis, drug resistance, and likelihood of metastasis development. It is therefore of great importance to identify appropriate therapeutic targets and gain a greater understanding of PDAC biology. Autophagy is a membrane-mediated degradation and recycling mechanism, which is crucial for cell homeostasis. There is evidence for both a tumor-suppressive and a tumor-promoting role of autophagy in cancer, and this is likely context dependent. Within PDAC, a large body of evidence points towards autophagy being required for tumor survival and metabolism. In this review, we describe the recent advances in the understanding of the role and regulation of autophagy in PDAC.

## 1. Introduction

Pancreatic ductal adenocarcinoma (PDAC) accounts for 90% of all diagnosed pancreatic cancers [1] and is listed as the seventh leading cause of cancer-related death in the world. A key challenge preventing successful PDAC treatment is the lack of early detection, which means that the majority of patients present with late stage surgically unresectable disease [2]. Another factor contributing to the poor prognosis of PDAC is therapeutic resistance to the current therapeutic options, traditional chemotherapy and radiation therapy.

General population screening is not a feasible strategy due to the low prevalence of PDAC, although higher-than-average risk groups have been identified, for example those with germline mutations or a history of pancreatitis [2]. Further progress is being made as traditional imaging technologies such as magnetic resonance imaging (MRI) and computer tomography (CT) are being complemented by new molecular imaging technologies such as positive emission tomography and hyperpolarized MRI [2]. Despite these advances, there is an unmet need for developing new therapies, which may include immunotherapy, targeted therapy or stroma-directed therapy. The development of such novel treatments requires an improved understanding of the molecular pathways involved in carcinogenesis and tumor progression, in which autophagy is thought to play a role.

There are several forms of autophagy: macroautophagy, referred to as autophagy, which involves the removal of cytoplasmic constituents and organelles, mitophagy, which refers to specifically mitochondria removal, microautophagy, which refers to molecules being directly invaginated by the lysosome, and chaperone-mediated autophagy [3]. Autophagy is a dynamic, highly conserved degradation and recycling mechanism for cellular components [4] which may be unnecessary or damaged. Breakdown products are subsequently used to sustain metabolic requirements. Autophagy is essential for homeostasis, cell metabolism and the preservation of organelle function and also has a function in viral and bacterial pathogen elimination. Autophagy is mediated by a unique organelle, the autophagosome, which surrounds a portion of the cytoplasm and delivers it to the lysosome for degradation [5]. The autophagosome membrane sources for this type of autophagy, as well as starvation-induced autophagy, are primarily the endoplasmic reticulum, but other membrane sources such as the Golgi, recycling endosomes and plasma membrane may also contribute [6]. The process is carried out by autophagy-related (ATG) proteins, coded for by at least 18 autophagy genes (*Atg* genes) [4,5]. Although there is low-level basal autophagy activity, the formation of the autophagosome is induced and controlled by factors such as starvation and stress. The initiation of autophagy is gradually becoming better understood, for example the role of the ATG9A protein in trafficking proteins and lipids to the autophagosome from the Golgi on autophagy induction [7]. Additionally, there is increasing evidence that lysosomes are also important in regulating the autophagic process, both through protein–protein interactions [8], and transcriptionally for example via TFEB [9]. As autophagy is such an important cellular process, it is controlled at multiple levels by a large number of signaling platforms located at specific membrane locations, such as the mitochondria and the nucleus [5].

The role of autophagy in PDAC is complex, with evidence pointing towards primarily towards a function in tumor cell survival—this has led to studies attempting to exploit autophagy as a therapeutic target. Primary pancreatic cancer tumors and cell lines show elevated autophagy levels under basal conditions, as measured by increased LC3-II expression (a membrane-associated marker for all stages of autophagy) and a greater number of autophagosomes per cell [10]. Autophagy inhibition via RNAi or small molecule inhibitors has been shown to cause death in PDAC cell lines and a reduction in tumor volume in PDAC xenograft models [11]. However, there are also indications that autophagy is dispensable for PDAC tumor growth [10,12] and clinical trials using autophagy inhibitors alone or in combination with other therapy have enjoyed limited success [13,14].

In this review, the evidence for the protective and tumorigenic role of autophagy in PDAC tumorigenesis will be summarized, followed by a description of recent advances in the understanding of how autophagy is regulated in PDAC.

## 2. Autophagy in Cancer

Autophagy in cancer has a complex context dependent role and has been associated with both a protective mechanism and cell death. A growing body of evidence has demonstrated that autophagy plays a part in nearly every phase of the metastatic cascade. This includes the initiation of tumorigenesis and cancer maintenance, as well as tumor cell invasion and motility, cancer stem cell differentiation and escape from immune surveillance [15]

Studies with genetically engineered mice have shown that autophagy suppresses primary tumor growth, whereas it is required for tumor maintenance and the progression to the advanced disease [15]. The initial evidence for the role of autophagy protecting against tumorigenesis was the study that demonstrated that *Beclin-1,* the mammalian autophagy gene, can inhibit tumorigenesis and is found at lower levels in human breast cancer [16]. Subsequent studies provided further evidence to support this observation, such as mice heterozygous for the autophagy gene *AMBRA-1* having increased rates of tumorigenesis [17].

On the other hand, it appears that cancer cells rely on autophagy for survival more than normal cells, and this reliance may further increase during therapy [3]. Autophagy is often upregulated in tumors, and, in solid cancers, such as breast and melanoma, increased LC3 puncta numbers positively correlate with a more aggressive phenotype [18]. Autophagy supplies metabolites to sustain the energy needs of the cancer cells and provides energy for malignant transformation [3]. In addition to autophagy induction by low nutrient and oxygen levels, autophagy can also be induced by high levels of reactive oxygen species (ROS), for example those generated by necroptosis, a regulated necrotic cell death process [19]. In some cases, however, an increase in ROS levels induces cell death in PDAC cell lines, which coincides with lower autophagy levels. For example, the membrane-permeable zinc-chelator TPEN reduces zinc availability, increases ROS levels, and decreases autophagy levels [20]. This highlights the complexity of autophagy regulation and the subsequent outcome of the pathway in PDAC cells.

A broad spectrum of cancers also rely on autophagy for survival in poorly oxygenated tumor areas, with data both in cell lines and xenograft models demonstrating that autophagy is required for cell survival under hypoxic conditions [21]. Autophagy may also prevent cell death by directly eliminating endogenous apoptosis inhibitors or creating an autophagosomal platform for caspase-8 activation [19]. A general theme that emerges from the study of autophagy in cancer is the existence of reciprocal relationships between autophagy and other cellular processes. For example, the interplay between autophagy and Rho activity is such that autophagy regulates Rho activity and Rho signaling modulates autophagy [15].

In order to enable the prediction of the outcome of any potential autophagy-related therapies, it is important to bear in mind not simply the role of autophagy in isolated tumor cells, but to also consider the tumor microenvironment. Particularly interesting in this regard is the role of autophagy in anticancer immunity, which is the recognition of antigenic peptides on cancer cells by T cells, leading to the elimination of the cancer cell [3]. There is evidence for both an inhibitory and an activating role of autophagy in this process. More specifically, autophagy has been shown to inhibit the formation of the immunological synapse and reduce cytolytic activity of T cells [22]. On the other hand, autophagy is essential for T cell and antigen-presenting cell function, which means that therapeutic autophagy inhibition is likely to affect the activity of intratumoral immune cells [3].

In conclusion, a wealth of evidence on the importance of autophagy in various types of cancer is available, but the complexity of the tumor environment makes it difficult to evaluate autophagy as a therapeutic target, as the different cell types within the tumor rely on autophagy to different extents.

## 3. Conflicting Roles of Autophagy in PDAC

Both in vivo and in vitro evidence points to a role of autophagy in maintaining the growth and survival of pancreatic cancer cells. However, even within the same study, the findings may point both to a pro- and an anti-survival role of autophagy. Although both PDAC primary tumors and cell lines show elevated basal autophagy [11], the exact role of the process is still to be clarified.

Both genetic and pharmacologic inhibition of autophagy has been shown to suppress growth of PDAC cell lines in vitro, possibly due to an increase in reactive oxygen species, elevated DNA damage, and decreased mitochondrial function [11]. Similarly, autophagy inhibition with shRNA targeting ATG5 or small molecule inhibitors leads to tumor regression and longer survival in pancreatic cancel xenografts and mouse models [11].

ATG5 and ATG7 proteins are essential for the formation of the autophagosome and the loss of the *Atg5* or *Atg7* genes abolishes autophagy [23]. A recent study has shown that pancreatic progression and tumorigenesis is impacted by dosage of the *Atg5* gene [24]. Here, the researchers generated mice that expressed oncogenic KRAS in pancreatic cells and had either homozygous, heterozygous or no disruption of *Atg5.* Unexpectedly, it was observed that reduced ATG5 protein levels promote tumor development, whereas homozygous disruption of Atg5 blocks tumorigenesis [24]. Primary pancreatic cancer cell lines in which Atg5 was knocked down showed an increased invasive and metastatic capacity when injected into mice. Human PDAC samples were also monitored and lower levels of Atg5 were associated with tumor metastasis and shorter survival time [24]. Monoallelic loss of *Atg5* led to resistance to the autophagy inhibitor chloroquine and a higher metastatic spread, which has important therapeutic implications, as chloroquine treatment may increase the risk of generating resistant cancer cell clones with heightened aggressiveness [24].

It is important to note that whether autophagy has a tumor-suppressive or a tumorigenic role is likely to be context dependent. Humanized genetically modified mouse models of PDAC lacking the essential autophagy genes *Atg5* or *Atg7* accumulate low-grade pre-malignant pancreatic lesions, suggesting that autophagy has a protective role and prevents tumorigenesis at its early stages [12]. However, in mice with oncogenic KRAS and a homozygous loss of P53, autophagy accelerates tumor onset rather than blocking tumor progression, with tumor growth fueled by the enhanced glucose uptake [12]. In addition, the loss of *Atg5* or *Atg7* in the pancreas with an activating *Kras* mutation has prevented the progression of premalignant lesions to invasive cancer [12]. A drawback of this study may be that a homozygous loss of P53 model was used, which may not be fully representative of human tumors where just one allele of P53 is lost [25].

An important feature of PDAC is the presence of therapy-resistant cancer stem cells, which are capable of self-renewal and migration [26]. A study comparing PDAC cells with high levels of stem cell markers demonstrated that cells with high levels had higher levels of autophagy [26]. Even more strikingly, autophagy inhibition resulted in the apoptotic cell death of PDAC stem cells and a reduction in their migratory activity and tumorigenicity [26]. Hypoxia has been shown to support invasive and stem-cell like characteristics of PDAC cell lines, as well as enhancing autophagy markers such as Beclin-1 and LC3-II [27]. This suggests that autophagy may promote the metastatic ability of cancer stem cells under the hypoxic conditions of the PDAC microenvironment. A later study then confirmed the connection between autophagy and PDAC stem cells by measuring stem cell markers such as aldehyde dehydrogenase (ALDH1) in patient pancreatic cancer tissue microarrays [28]. A high co-expression of the autophagy markers LC3-II and ALDH1 in patients correlated with poor survival and decreased progression-free survival [28]. Confirming these observations, cell line experiments revealed that sphere-forming stem cell-like PDAC cells have higher levels of the autophagy marker LC3-II compared to other cells [28]. Another link between autophagy and PDAC cancer stem cells is YAP/TAZ-signaling, shown to be required for cell dedifferentiation and the acquisition of self-renewing properties, i.e., stem cell formation [29]. YAP/TAZ has also been shown to be a positive autophagy regulator [30], and promotes autophagic flux by increasing Armus expression, a RAB7-GAP required for autophagosome turnover [29]. The role of autophagy in PDAC survival is not limited to tumor cells alone, but extends to the tumor microenvironment. Stroma-associated pancreatic stellate cells secrete non-essential amino acids needed for PDAC metabolism [31]. Autophagy in stellate cells is stimulated by cancer cells and required to enable stellate stroma cells to secrete alanine, which then acts as an alternative carbon source for cancer cells [31]. This fuel shift is important for PDAC tumors, which are characterized by a microenvironment limited in glucose and serum-derived nutrients. Similar findings have been made in lung adenocarcinoma and other cancers [32].

However, there is also evidence to the contrary, showing that autophagy can play a role in protecting against tumorigenesis in the pancreas. For example, autophagy has been shown to prevent endoplasmic reticulum (ER) stress in pancreatic acinar cells, exocrine pancreas cells which produce and secrete large amounts of digestive enzymes [33]. This enzyme secretion function of acinar cells means they have an extensive ER network, and ER stress has been shown to contribute to pancreatitis, a risk factor for PDAC development [34]. Mice lacking ATG7, which is essential for autophagy, show reduced autophagic flux, and a concomitant increase in ER stress, acinar cell degeneration and pancreatic inflammation [33]. Furthermore, in pancreatic **β** cells, autophagy may impact development and initiation. Autophagy in NIT-1 mouse pancreatic **β**-cells and in vivo in mice has been shown to be activated by kisspeptin, which is overexpressed in type II diabetes, a critical PDAC risk factor [35].

## 4. Recent Advances in the Understanding of the Control of Autophagy in PDAC

Autophagy is a highly regulated process at a range of levels, starting from transcriptional control to epigenetic and post-translational mechanisms. Given the importance of autophagy in PDAC, as summarized in the previous section, the control of autophagy has been widely studied, and has been reviewed previously [36]. The next section of this review summarizes the recent advances in the understanding of how autophagy is regulated in PDAC, by discussing the new regulators recently uncovered and, where possible, explaining how they may function in autophagy regulation. These new regulators are summarized in Figure 1, which highlights the complexity and extent of autophagy regulation in PDAC1.

### 4.1. Positive Regulators of Autophagy in PDAC

#### 4.1.1. KRAS

Overall, 95% of cases of PDAC have *Kras* mutations [37], and the current anti-KRAS therapy research is following several directions. In order to understand whether the loss of KRAS drives high basal autophagy levels seen in *KRAS* mutant PDAC, KRAS was acutely suppressed in a panel of human and mouse PDAC cell lines using siRNA and small molecule inhibitors of the KRAS effector ERK MAPK [38]. Unexpectedly, KRAS suppression and ERK inhibition resulted in a greater autophagic flux, and a decrease in glycolytic and mitochondrial function. This indicates that PDAC dependence on autophagy is increased by ERK inhibition, and pharmacologic inhibitors that act against both ERK MAPK and autophagy are required for effective PDAC treatment [38]. This is supported by experiments where the autophagy inhibitor chloroquine synergistically improved ERK inhibitor anti-tumor activity [38]. Additionally, a study where siRNA was used to identify the dependency patterns in *KRAS* mutant and WT pancreatic cancer cell lines showed that targeting BRAF and CRAF kinases in combination with the autophagy E1 ligase ATG7 successfully eliminated *KRAS* mutant cells, while keeping toxicity in healthy cells to a minimum [39] There is, therefore, substantial evidence from a variety of sources indicating that MAP kinase and autophagy pathways work together to maintain RAS mutant tumor survival.

One of the ways in which oncogenic KRAS positively regulates autophagy in PDAC may be through the induction of VMP1 (Vacuole membrane protein 1), which is needed for autophagosome formation [40]. RNAi experiments have shown that VMP1 is needed for KRAS to induce and maintain autophagy, and this process involves GLI3, a transcription factor regulated by the Hedgehog pathway, which also induces the transcription of VMP1 [40]. Furthermore, electron microscopy ultrastructural analysis showed that cells deficient in VMP1 were able to generate small premature autophagosome-like structures, which could not elongate or mature into autophagosomes [41].

#### 4.1.2. Osteopontin

Osteopontin (OPN) is a secreted glycoprotein with roles in cancer progression and autophagy induction in smooth muscle cells through the integrin/CD44- and p38 MAPK-mediated pathways [42]. A study investigating PDAC stem cells has shown that osteopontin (OPN) stimulates LC3-II protein expression and therefore increases the LC3-II/LC3-I ratio, indicating higher autophagic flux, as well as stimulating stem cell markers such as ALDH1 and CD44 [28]. OPN-induced autophagy could be prevented by the inhibition of NF-kB activation, but not the inhibition of other signaling pathways downstream of OPN [42].

#### 4.1.3. SNHG14

Gemcitabine is one of the conventional chemotherapeutic agents used to treat PDAC, but patients often have a limited response due to resistance [13]. Autophagy inhibition has been shown to reduce pancreatic stem cell numbers and their ability to form spheres [28]. Autophagy inhibition also increased the sensitivity of PDAC cells to gemcitabine [28]. Long non-coding RNAs (RNAs more than 200 nucleotides in length) may regulate autophagy in cancer cells [43]. Small nucleolar RNA host gene 14 (SNHG14) in particular has a role in promoting cancer progression in several cancer types, and is expressed more highly in PDAC compared to normal tissue [44]. SNHG-14 has been shown to interact with the microRNA miR-101, a negative autophagy regulator. This interaction could lead to reduced miR-101 levels, and therefore promote autophagy and increase the resistance of PDAC cells to gemcitabine [44,45].

### 4.2. Negative Regulators of Autophagy in PDAC

#### 4.2.1. UBL4A

Lysosome-associated membrane protein-1 (LAMP1) and LAMP2 are regulators of autophagosome maturation and the major components of the lysosomal membrane [23]. Other evidence also suggests that metabolic changes in PDAC drive the transcriptional activation of lysosome biogenesis [46]. A recent study has identified ubiquitin-like protein 4A (UBL4A), a tumor suppressor mediating cell death in response to DNA damage [47], and a protein folding chaperone [48], to directly interact with LAMP1 [49]. The interaction between UBL4A and LAMP1 is thought to disturb lysosome function and thus impair autophagic degradation, as observed by the autolysosome accumulation and lysosomal disfunction in cells with higher UBL4A levels [49]. Furthermore, the analysis of UBL4A expression in 69 PDAC patients revealed that patients with higher levels of UBL4A mRNA in PDAC tissue have improved survival rates [49], perhaps due to also having lower autophagy levels.

#### 4.2.2. Optineurin

Optineurin is a membrane trafficking protein with a role in selective autophagy as a cargo receptor, whereby it delivers polyubiquitinated cargo to the autophagosome through its LC3-interacting region [45]. The Human Protein Atlas shows optineurin to be overexpressed in pancreatic cancer, along with other autophagy proteins such as LC3 and GABARAPL2 [50]. A panel of PDAC cell lines was used to probe optineurin function, revealing that, while siRNA depletion of optineurin does not significantly impact cell survival, it does reduce the ability of cells to undergo colony formation and increases the subG1 (pre-G1, a cell death indicator) proportion of cells [50]. Interestingly, these data show that optineurin knockdown only mildly affected macroautophagy levels, with a predominant effect on chaperone mediated autophagy [50].

#### 4.2.3. proNGF

Precursor of nerve growth factor (proNGF), a regulator of neuronal regeneration and development, is found to be overexpressed in a range of malignant tumor cells [51]. The siRNA-mediated depletion of proNGF in PDAC cell lines results in a reduction of the level of ATG5 and BECN1, together with an increase in P62 levels, a cargo protein which is degraded by autophagy, suggesting that proNGF knockdown may suppress autophagy [51]. ProNGF was also shown to be required for PDAC cell proliferation, migration and invasiveness [51]. However, a limitation of this study was that only siRNA was used to modulate proNGF levels. Interestingly, proNGF was also found to be upregulated in PDAC samples compared to paracancerous tissues in 60 patients [51].

## 5. Autophagy as a Therapeutic Target for Pancreatic Cancer

The strong evidence for autophagy playing a role in pancreatic cancer survival and tumorigenesis has led to autophagy being considered a promising therapeutic target, although the clinical outcomes show complexity and could be considered disappointing. For example, a recent randomized phase II trial in patients with metastatic pancreatic cancer of gemcitabine/nab-paclitaxel with or without the hydroxychloroquine (HCQ), an autophagy inhibitor, has shown that the partial response is improved in patients receiving HCQ treatment as well as gemcitabine/nab-paclitaxel (46% partial response) compared to gemcitabine/nab-paclitaxel alone (17%). However, both 1-year survival and overall survival were shorter in the HCQ treated patients [13]. Earlier attempts to use HCQ, a derivative of chloroquine, as a monotherapy in patients with metastatic pancreatic adenocarcinoma were similarly unsuccessful and showed negative efficacy [14]. This may be explained by the doses of HCQ used in the study being insufficient to inhibit autophagy, as measured by LC3-II levels [14].

As discussed above, the autophagy inhibitor HCQ or chloroquine (CQ) has shown limited efficacy in PDAC, either as a monotherapy or when combined with standard-of-care therapies. In order to identify the potential molecules which may be combined with CQ treatment for improved clinical efficacy, an unbiased pharmacological inhibition kinome screen in PDAC cells has been performed, which revealed that replication stress inhibitors were synthetically lethal with CQ [52]. One of these inhibitors, the ATR inhibitor VE-822, was selected for further analysis due to its translational potential, with its favorable pharmacodynamics and good tolerance in animal models [53]. This inhibitor showed synergy with CQ in a number of PDAC models, including 2D and 3D cultures, as well as an in vivo xenograft model [52].

Combination therapy where hydroxychloroquine is combined with MEK1/2 inhibition makes sense biologically, as PDAC is a RAS-driven cancer and MEK1/2 inhibition activates autophagy. This combination treatment has shown promising results in patient-derived xenografts in mice, PDAC cell lines and even a partial response in a patient [54]. In addition to the possibilities inherent in combination therapy, pre-clinical studies continue to provide us with new chemical moieties which are potentially promising in PDAC therapy. These include metal moeities such as oxidovanadium (IV) coordination complexes which have been shown to be toxic to PDAC cell lines, but not non-tumor human immortalized pancreas duct epithelial cells [55]. Interestingly, these compounds induce, rather than inhibit autophagy, so their anti-proliferative activity may be exerted via other mechanisms, such as increased reactive oxygen species generation [55].

It is important to note that the type of treatment that proves most beneficial may be specific to a subtype of PDAC, for example PDAC cells overexpressing insulin-like growth factor have been shown to respond to treatment with small molecule heat shock protein 90 inhibitors, perhaps via ERK signaling inhibition [56].

Studies in cancers other than PDAC may provide the field with additional therapeutic targets, particularly if there is a link in the mechanisms that drive these cancers and PDAC. For example, novel non-toxic inhibitors of γ-Glutamyl transpeptidase (GGT) are under development and have been shown to induce cell death in liver cancer and leukemic cells overexpressing GGT [57]. This is of relevance to PDAC treatment, because of the role of GGT is glutathione metabolism and in autophagy, processes which PDAC cells rely on for survival [58].

## 6. Conclusions and Future Directions

Autophagy has a complex context-dependent role in cancer, with evidence suggesting both a protective role during early stages of tumorigenesis, and a role in promoting tumor survival in several cancer types. For PDAC, the majority of studies indicate in metastatic tumors a pro-tumor survival role for autophagy, achieved largely by the contribution of energy and intermediates such as alanine. However, despite the promising data in PDAC cell lines and mouse xenografts showing that autophagy inhibition reduces cell proliferation and tumor size and prolongs mouse survival [11,26], clinical outcomes in autophagy inhibitor trials have not seen an improvement on standard-of-care treatment.

Given the recent evidence suggesting that the level of ATG5 may determine the outcome of autophagy inhibition treatment [24], it would be of interest to perform a correlative study between patient response to pharmacological autophagy inhibition and ATG5 levels, alongside other proteins involved in autophagy, in the hope of finding biomarkers for treatment.

Another potential direction could be to monitor the degree to which autophagy has been inhibited and see if this differs in patient cohorts. One possible approach here could include the use of pharmacodynamic markers to assess the extent to which autophagy inhibition has been successful. One example of such a marker is LC3-II, monitored in human peripheral lymphocytes after HCQ treatment [14]. It is important to ensure that autophagy in tumors is adequately inhibited. It might be that autophagy inhibition alone is insufficient to affect pancreatic tumor growth, and in this case combination treatments will likely be explored to elicit a therapeutic response.

## Figures and Tables

**Figure 1 biology-09-00007-f001:**
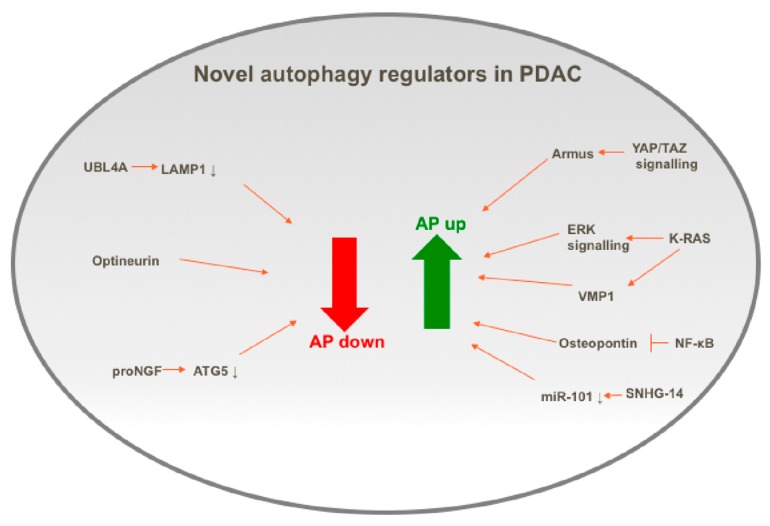
Pictorial summary of positive and negative regulators of autophagy (AP) in PDAC cells.

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
