# Peer review of "The Role of Autophagy in Pancreatic Cancer—Recent Advances"

_biology, 2019, doi:10.3390/biology9010007_

Round 1
Reviewer 1 Report
The review article by Maria New and Sharon Tooze deals with the role of autophagy in pancreatic cancer highlighting the conflicting action in both tumor suppression and tumor progression.
The review is clear and well-written and displays interesting aspects for cancer community.
I have only a comment/curiosity about the recent evidence suggested by Brancaccio et al., about the interplay of glutathione metabolism and amino acid recycling in some tumours and autophagy process (J Biol Chem. 2019 294(40):14603-14614. doi: 10.1074/jbc.RA119.009304.) I am wondering if also some pancreatic tumours can express foci of GGT over-expression and be subject to an autophagic regulation thereof. This aspect could be nice to discuss.
Author Response
In response to reviewer 1, we have added a section in the “5. Autophagy as a therapeutic target for pancreatic cancer” section to discuss the possibility of GGT treatment in PDAC (highlighted in yellow). Although there is no published data showing GGT to be overexpressed in PDAC cells, it’s role in glutathione metabolism indicates that it may be a potential therapeutic target to investigate, as glutathione levels play a key role in PDAC survival. This is in addition to the role of GGT in autophagy, which is crucial in PDAC, as discussed in the review article presented.
Reviewer 2 Report
The Authors provide a comprehensive systematic overview of the research in the field.
The following reports should be discussed:
Nat Med. 2019 Apr;25(4):620-627.
Sci Rep. 2019 Nov 25;9(1):17451. doi: 10.1038/s41598-019-53826-7.
Int J Mol Sci. 2019 Jan 10;20(2).
Proc Natl Acad Sci U S A. 2019 Sep 3;116(36):17848-17857.
Proc Natl Acad Sci U S A. 2019 Mar 5;116(10):4508-4517.
Mol Cancer. 2019 May 23;18(1):100.
J Cell Physiol. 2019 Nov;234(11):20648-20661.
Cell Mol Life Sci. 2019 Sep;76(17):3433-3447.
The Authors have to incorporate a pictorial or cartoon representation summarizing the main topics addressed in the review.
Author Response
In response to reviewer 2 we have added key messages of each of the additional suggested 8 references to relevant sections of our article (highlighted in green). We feel these recent publications are valuable additions to the review. We have also added a cartoon representation summarising the main topics of the review – Figure 1. The text for the addition of Figure 1 is highlighted in green.
Round 2
Reviewer 2 Report
No further comments